# Using a Fiber Bragg Grating Sensor to Measure Residual Strain in the Vacuum-Assisted Resin Transfer Molding Process

**DOI:** 10.3390/polym14071446

**Published:** 2022-04-01

**Authors:** Guang-Min Luo, Guang-Yen Liou, Hong-Zhe Xiao

**Affiliations:** Department of Naval Architecture and Ocean Engineering, National Kaohsiung University of Science and Technology, Kaohsiung 811532, Taiwan; f108186105@nkust.edu.tw (G.-Y.L.); f109186101@nkust.edu.tw (H.-Z.X.)

**Keywords:** FRP, residual strain, FBG sensor, VARTM

## Abstract

Vinyl ester (VE) resin has strong environmental tolerance and is the matrix commonly used in the composite materials of fiber-reinforced plastics (FRP). VE resin is often combined with glass fiber in different maritime structures, such as wind turbine blades, spinner cases, and nacelle cases. However, VE resin exhibits exothermic reactions and shrinkage during curing, which often generates residual strain in large structures and those with a high stacking number. This study explored the exothermic reaction and shrinkage of VE resin and glass fiber during the vacuum-assisted resin transfer molding process, as measured using a fiber Bragg grating sensor. The experiment results verified the relationship between the stacking number and residual strain shrinkage. In addition, the symmetric laminate method was used to prevent the bending–twisting coupling effect and subsequent warping deformation of the FRP laminated plate during curing. The experiment results also verified that the bottom layers of the FRP laminated plates produced using VE resin were closer to the mold, and exhibited more shrinkage as the stacking number increased. In addition, this study discovered that during the experiment, the symmetry layer of the FRP laminated plate had a higher exothermic temperature than the bottom layer as a result of the symmetry layer’s ineffective heat dissipation. Therefore, the curing shrinkage of the symmetry layer resin was measured. The experiment results indicated that if the stacking number was between 10 and 30, the residual strain shrinkage of the symmetry layer was greater than that of the surface layer. However, because of the symmetric laminate, the residual strain of the symmetry layer did not increase when the temperature increased. Therefore, the greatest residual strain occurred at the surface of the bottom layer of the laminated plate with a stacking number of 40.

## 1. Introduction

The vacuum-assisted resin transfer molding (VARTM) process is a common method used to produce large fiber-reinforced plastic (FRP) structures. This process controls the resin content and produces reliable FRP products and has been widely used in the production of yacht and warship superstructures and wind turbine blades. For harsh maritime environments, FRP marine structures often employ vinyl ester (VE) resin as the matrix on account of its strong environmental tolerance. However, the curing of VE resin is exothermic, and considerable shrinkage occurs after the curing process. With FRP marine structures increasing in size, their production requires a higher stacking number, resulting in more residual strain induced through the exothermic reactions and shrinkage of VE resin after curing. This study used fiber Bragg grating (FBG) monitoring technology to measure the exothermic reactions during curing and residual strain shrinkage after curing of FRP laminated plates of different stacking numbers, verifying that FBG sensors can be applied in the monitoring of the VARTM process.

The FRP laminated plates employed in this study were produced using the VARTM process. The raw materials were LT-800/M-225 glass cloth and VE resin, and an FBG sensor was used to monitor the curing process of the FRP laminated plates. The diameter of the FBG sensor is smaller than that of glass fiber; hence, burying it into glass cloths does not affect the mechanical characteristics of the cloth. After impregnation of the glass cloth and VE resin, the FBG sensor combined with the resin was used to record the temperature and volume changes of the FRP laminated plate during curing of the resin.

FBG was discovered by Hill et al. [1] in 1978; they used the mutual interference of the incident wave and partially reflected wave of laser beams to create standing waves in optical fibers, forming a periodical grating structure. In 1993, Hill [2] applied an excimer laser and phase mask and used diffraction to draw the Bragg grating with a specific period on the optical fiber. Currently, the phase mask is the most common method for producing Bragg grating and was also used in this study to produce the FBG sensor.

Since its development in 1993, FBG has been used in many fields. Rao [3] conducted a detailed review of the development of FBG, verifying that its measurement precision can reach 1 ± 0.5 με. Pereira [4] employed FBG without coating to perform strain measurement and compared the results with those of the strain gauge. The measurement precision of FBG without coating reached 0.1 με, and the FBG exhibited superior linear performance compared with that of the strain gauge. In 2016, Alwis [5] reviewed the developments and applications of optical fiber measurements over the past 20 years and reported that current optical fiber sensors could perform multichannel measurements. Di [6] demonstrated that optical fiber sensors can be applied in harsh and high-temperature measurement environments, and are suitable for monitoring both the exothermic reaction during VARTM and residual strain between layers. Min [7] showed the feasibility of using optical fiber sensing technology for marine application and it is possible to envisage a widespread use in this research field in the next few years.

FBG can be used to measure physical quantities such as temperature, strain, and vibration. In terms of temperature monitoring, Zhan [8] designed an FBG sensor that could operate under high temperatures and experimentally verified that the FBG sensor could function and monitor normally between 0 °C and 800 °C. Rajan [9] buried FBG sensors made from different materials into an FRP laminated plate, which was then heated to explore the strain and temperature sensitivity of the different FBG sensors. Woyessa [10] discovered that exerting prestrain on FBG sensors under room temperature can increase their sensitivity in temperature and pressure monitoring. However, when the temperature exceeds 35 °C, the prestrain is cancelled out through the thermal expansion of the optical fiber, and exerting prestrain will not improve the sensitivity of the FBG sensors.

In addition to temperature monitoring, FBG sensors can also be used to measure the changes of the strain field. Dewynter-Marty [11] was the first to bury a FBG sensor with a measurement precision of 1.01 με into a glass fiber laminated plate, demonstrating that the glass fiber plate and laminated plate with epoxy resin have a residual strain of 2200 με after curing. Moreover, Kuang [12] buried carbon fiber, glass fiber, and metallic fiber into FBG separately to conduct measurements and reported that the strain fields of the asymmetric laminates in FBG often cause the birefringence effect, leading to a split in the monitoring frequency spectrum. Kang [13,14] combined FBG and external Fabry–Perot interference to create a new sensor, which was then buried in a uniaxial and cross-ply carbon-fiber-reinforced polymer laminated plate to measure the residual strain of the asymmetric and symmetric laminates. The results indicated that FBG sensors achieve an accurate linear performance in temperature measurements.

FBG sensors have been used in studies related to the monitoring of the curing of FRP resin for over 20 years. Chehura [15] monitored the curing behavior of epoxy resin impregnated with glass fiber, with the results indicating that the epoxy resin with vertical fibers exhibited a greater shrinkage strain after curing. Frazao [16] buried three single-mode optical fibers in laminates mixed with carbon fiber and glass fiber and observed that, owing to the different thermal expansion coefficient of glass fiber and carbon fiber, their interfaces exhibited evident deformation and extrusion during thermoforming. Dai [17] used FBG sensors to measure the heat transfer and shrinkage of a carbon-fiber–reinforced laminated plate during curing to construct a numerical model that could simulate the residual stress of curing. Furthermore, Kang [18] buried FBG sensors with different grating lengths into glass fiber and carbon fiber, revealing that shorter grating was less likely to induce the birefringence effect, and can reduce the effect of the lateral force produced during the curing of resin.

The usage of FBG sensors in monitor the production of composite materials has been continually developed and refined, increasingly accounting for the fundamental characteristics of materials and their mechanical behaviors. In terms of the fundamental characteristics, Hu [19] employed FBG sensors to monitor the shrinkage rate of composite materials and successfully obtained the gel point of resin. Hoffman [20] explored the gel point of epoxy resin during curing and the time required for complete curing, using FBG sensor and thermocouple to collect measurements. The study successfully obtained the curing curve of epoxy resin, with the experiment results verifying the applicability of the theoretical equation. When the essence of resin changes, the curing of epoxy resin does not require high temperatures; therefore, Khadka [21] used FBG sensors and thermocouples to measure the shrinkage strain of epoxy resin cured under high temperatures and room temperature. Regarding the measurement of mechanical behaviors, Zanjani [22] applied four sets of FBG sensors to monitor the asymmetrical deformation behavior of carbon fiber, glass fiber, and epoxy composite during the heating process. In addition, to monitor the tensile strength of glass fiber, Chen [23] impregnated both glass fiber and an FBG sensor with epoxy resin and used pultrusion to manufacture an FRP bar. The study results verified that the tensile strength of fiber is 15.3 times higher than its compressible strength. In addition to the monitoring of strength and deformation, FBG sensors can be used to monitor the delamination crack propagation in FRP. Kakei [24] experimentally demonstrated that the delamination crack propagation caused nonuniform distortion in the strain signal of FBG sensors. Thus, nonlinear distortions can be used to determine if a delamination crack has occurred.

During the curing shrinkage of resin, the temperature remaining in the laminated plate affects the accuracy of the measured shrinkage rate of the resin. Therefore, when using an FBG sensor to measure the residual strain of resin after curing, these temperature effects must be corrected. To measure the resin shrinkage during thermoforming, Boateng [25] designed different insulation envelopes to envelope the FBG sensor and insulate heat, ensuring that the FBG sensor would only detect the shrinkage of the resin. The design of the insulation envelopes could also be used in the measurement of heterogeneous residual strain; Huang [26] successfully used FBG and insulation envelopes to monitor the residual strain of a heterogeneous bonding structure. Alcock [27] used binding agent and a guide-tube to embed FBG sensors into the Li-ion Battery (LIB) surface and monitor the continuous temperature. Yazd [28] used a three-variable two-level factorial design to assess fiber Bragg grating properties under simultaneous temperature, humidity, and strain stimuli. More importantly, the technology Yard developed quantifies the cross-sensitivities between temperature and strain, temperature and humidity, humidity, and strain, and between all three factors. To clearly present the overall phenomenon of the curing of VE resin, insulation envelopes were not employed in this study. Instead, the method of Liang [29] was referenced, and the temperature and strain sensitivity coefficient were used to immediately correct the effect of the temperature on the shrinkage strain.

In conclusion, this study verified that FBG sensors can be used to monitor the curing reactions of multilayered and highly exothermic FRP laminated plates. In terms of the measurement of the curing shrinkage of resin, an initial experiment was conducted to obtain the strain sensitivity and temperature sensitivity coefficients of the FBG sensor, and the coefficients were used to correct the temperature compensation and prevent distortion of the temperature measurements. Finally, this study obtained the residual strain of VE resin during the production of a thick FRP structure using the VARTM process; the residual strain corresponded to the increasing stacking number of the VE resin.

## 2. Research Method

### 2.1. Relationship between the Wavelength Shift and Strain

This study used a single-mode optical fiber with a diameter of 245 μm and phase mask to manufacture the FBG sensor used for measuring, with wavelength modulation employed to measure the changes of the physical quantities. The optical fiber used in this study was Corning SMF-28e, and the coating diameter was 245 μm. The FBG sensor of the wavelength modulation mainly relied on the grating distance changes to create the shift of the reflected wavelength. Obtaining the relationship between the wavelength shift and grating distance change enabled the use of the wavelength shift to obtain the corresponding strain.

We used the phase grating method to scribe the FBG sensor in the experiment, as depicted in Figure 1. First, we placed the optical fiber under the phase mask at which the ultraviolet laser was directed. The laser passing through the grating generated mutual interference resulting from diffraction, forming a permanent grating structure in the optical fiber. The distance between the gratings were expected to affect the test results. Therefore, to ensure the stability of the FBG sensor, we used an optical spectrum analyzer and broadband amplified spontaneous emission light source to test the center wavelength of all FBG sensors. The OSA used in this study is Anritsu MS9710C, the measurement range of spectral wavelength is 600~1750 nm, and the maximum spectral resolution can be accurate to 0.05 nm. In order to prevent the waveforms measured by each stacking layer from overlapping, we adjusted the spacing of phase mask to obtain different center wavelengths. The effective refractive index of optical fiber used in this study was 1.468, and the spacing of phase mask considered in this study was between 0.521~0.531 μm. Finally, we obtained an optical fiber with a center wavelength of 1530~1560 nm.

The test demonstrated that the center-reflected wavelength of the FBG sensor used in this study was between 1530 and 1550 nm. To validate the relationship between the wavelength shift and strain of the FBG sensor used in this study, a tensile test was conducted. First, we used ASTM D3039 to fabricate the FRP tensile test piece and then pasted the FBG sensor and strain gauge onto the test piece, as illustrated in Figure 2. Next, we conducted a tensile test and recorded the wavelength shift and strain; the elongation rate was set at 2 mm/min. In addition, the fracture strain of the FRP was approximately 2%. To ensure that the recorded strain had a linear change, we only retrieved data within 6 mm of elongation; the test results are summarized in Table 1. According to the results, the relationship between the wavelength shift and strain was stable at different time points. A wavelength shift of 1 nm of the FBG sensor corresponded to an average strain of 0.00075. Therefore, the wavelength shift corresponding to 1 με was 0.0013 nm, which is close to the test results of Frazao.

### 2.2. Temperature and Strain Sensitivity Coefficient

The curing and shrinkage of VE resin are accompanied by high exothermic behaviors. Therefore, this study used the FBG sensor to measure the residual strain of VE resin after curing, correcting for the effect of the temperature. Liang [26] applied the strain sensitivity coefficient $K_{\varepsilon }$ and temperature sensitivity coefficient $K_{T}$ to represent the wavelength shift Δλ affected by the temperature and strain field.
(1)$$\frac{\Delta \lambda }{\lambda_{B}}=K_{\varepsilon }\Delta \varepsilon +K_{T}\Delta T$$

In Equation (1), $\lambda_{B}$ is the center wavelength of the FBG sensor and Δε and ΔT are the change of the strain and temperature, respectively. With measurement using the FBG sensor, Δλ is a known condition, and the change of the temperature can be monitored in real time using the thermocouple. Therefore, if $K_{\varepsilon }$ and $K_{T}$ can be obtained, then the effect of the temperature on the strain can be corrected, and the real strain Δε of the resin shrinkage can be obtained.
(2)$$\Delta \varepsilon =\frac{1}{K_{\varepsilon }}\left(\frac{\Delta \lambda }{\lambda_{B}}-K_{T}\Delta T\right)$$

Vertical strain tests were performed to obtain $K_{\varepsilon }$, as expressed in Equation (3).
(3)$$K_{\varepsilon }=\frac{1}{\lambda_{B}}\times \frac{\Delta \lambda }{\Delta \varepsilon }$$

This study used a phase mask to fabricate the FBG sensor, and the center wavelength $\lambda_{B}$ thus exhibited some differences; the wavelength was approximately 1540 nm. In Equation (3), $\lambda_{B}$ was assumed to be 1540 nm. In addition, the relationship between Δλ and Δε was obtained through the use of determined relationship between the strain and wavelength shift as a reference. This relationship was substituted into Equation (3) to obtain $K_{\varepsilon }=8.44\times 10^{-7}$.

When the FBG sensor was used to measure the temperature change, the temperature sensitivity coefficient $K_{T}$ presented in Equation (4) can be used if the effect of the temperature on the wavelength shift is the only effect considered.
(4)$$K_{T}=\frac{1}{\lambda_{B}}\times \frac{\Delta \lambda }{\Delta T}$$

In Equation (4), $\lambda_{B}$ was assumed to be 1540 nm; however, the wavelength shift and temperature change must be obtained through additional heating experiments. We designed a fixture and attached the FBG sensor and thermocouple to the fixture, which was then placed in an oven for four separate heating tests, as depicted in Figure 3. During the heating tests, we straightened the FBG sensor to avoid curving to cause errors in the measurements. Additionally, a set of thermocouple sensors was placed on both ends of the FBG sensor to verify the consistency of the temperature distribution around the FBG sensor. This test was performed to obtain the temperature sensitivity coefficient $K_{T}$ of the FBG sensor, as control of the temperature was crucial. We used a programmable oven and began the measurement of the wavelength shift from 30 °C to 60 °C, recording a measurement every time the temperature rose by 5 °C. To ensure temperature stability, we maintained the same temperature for 10 min every time the temperature rose by 5 °C. After the wavelength shift was recorded, the temperature was increased to the next predetermined temperature.

The four heating experiments yielded similar results. When the temperature rose by 1 °C, the wavelength shift was approximately 0.01 nm. Therefore, we used Equation (4) to verify that the FBG sensor in this study had a $K_{T}=7.03\times 10^{-6}$. Figure 4 illustrates the wavelength shift Δλ and temperature change ΔT curve results from two of the heating tests.

### 2.3. Lamination Conditions and Sensor Placement

This study employed VARTM to produce the FRP laminated specimens and a resin composed of VE resin with LT-800/M-225 glass cloth. The stacking numbers of the FRP laminated specimens were 4, 10, 20, 30, and 40 layers. To prevent asymmetrical deformation during the curing of the test piece, this study used the symmetrical laminate method to lay the glass cloth.

Production experience has indicated that, as a result of heat dissipation problems and boundary restrictions, FRP laminated plates produced using VARTM often exhibit shrinkage or warping deformation on the side closer to the mold. Therefore, we expected the surface layer of the laminated plate to have the greatest residual strain shrinkage and thus buried the FBG sensor in the surface layer to perform measurements. In addition, during the monitoring of the curing temperature of the FRP laminated plate, we discovered that the symmetry layers did not dissipate heat easily, with the area exhibiting the highest temperature. For example, if the stacking number was 40 layers, then the maximum exothermic temperature of the symmetry layer of the laminated plate could reach 130 °C, which was at least 30 °C higher than that of the bottom layers closer to the mold. Therefore, we also buried FBG in the symmetry layers to observe the effect of the exothermic reaction during the curing of the resin on the final residual strain.

Figure 5 depicts the symmetrical laminate method used in this study as well as the placement of the thermocouple and FBG sensor. The FBG sensor used to measure the bottom layer residual strain of the laminated plate was placed between the first and second stacking layers, and the thermocouple sensor was placed close to the FBG sensor to monitor temperature changes for future temperature correction. In addition, we placed another FBG sensor and thermocouple set in the symmetry laminate to simultaneously monitor the temperature and residual strain of the symmetry layer of the laminated plate.

The VARTM process uses negative pressure to cause the resin to flow and complete infusion. To prevent the resin from pushing and moving the FBG sensor while it flowed and causing errors in the measurements, the FBG sensor was fixed in place with glue. In addition, the LT-800/M-225 glass cloth had a groove at 0 degrees, as presented in Figure 6b. Therefore, we placed the FBG sensor and thermocouple within the groove, ensuring that the FBG sensor was parallel to the flow of the resin infusion to prevent the resin infusion from exerting lateral forces on the FBG sensor, causing it to displace or bend, which could affect the precision of the measurement. Figure 6 depicts the placement of the FBG sensor and use of VARTM to infuse the FRP laminated plate.

## 3. Measurement Results and Discussion

### 3.1. Explanation of the Measurement Results of the FBG Sensor

All measurements in this study were collected in a laboratory under controlled temperature and humidity. The temperature during the infusion of the different layers of the laminated plates ranged from 31.2 °C to 31.8 ℃, and the relative humidity was between 71% and 73%; thus, environmental effects were minimized. In this section, the measurement results of the bottom layer with a stacking number of 30 are employed as an example to explain the results. Real-time monitoring was performed throughout the experiment, with the wavelength shift of the FRP laminated plate during resin infusion, exothermic reaction, and shrinkage measured. Next, the wavelength shift was used to estimate the shrinkage and residual strain of the FRP laminated plate.

When using VARTM to fabricate the FRP test piece, the test piece must be vacuumed, and pressure holding applied. Resin infusion can begin only after the vacuum level has been verified. Figure 7 presents the real-time measurements of the laminated plate with a stacking number of 30 during vacuuming, resin infusion, and exothermic reaction and complete curing. The “initial condition” illustrated in Figure 7 refers to the center wavelength of the FBG sensor without loading. After we began the vacuuming, the FBG sensor and glass cloth were placed under negative pressure causing extrusion. When the FBG sensor was placed under a compressive load, the center wavelength shifted toward the short wavelength. When we applied pressure holding, the wavelength slowly increased to 1529.178 nm.

During resin infusion, the environmental pressure slowly increased back to one atmosphere pressure, and the center wavelength of the FBG sensor returned to the “initial condition.” After this, the resin exhibited the curing reaction and dissipated large amounts of heat, resulting in the thermal expansion of the FBG sensor, and the rapid shifting of the center wavelength toward the long wavelength. According to the test results of the gelation test, the curing agent ratio of this study allowed the VE resin to complete curing within 30 to 40 min, followed by the slow cool down and shrinkage of the resin.

Figure 7 illustrates the real-time wavelength shift, though the obtained curve does not reveal the relationship between the wavelength shift and temperature changes between the FRP layers. Therefore, we employed the temperature data measured using the thermocouple as the abscissa and used the wavelength shift to represent the relationship between the wavelength shift and temperature changes between the FRP layers (Figure 8). Regarding the wavelength data, simple maximum value of optical spectrum is directly taken, and the sampling frequency is 30 s.

As depicted in Figure 8, the temperature of the FRP laminated plate did not markedly change during vacuuming and resin infusion and remained similar to the environmental temperature (31.2 °C). When the curing process of the resin began, the surface layer closer to the mold exhibited the highest exothermic temperature (87 °C), and the wavelength of the FBG sensor began to shift toward long wavelength. Following completion of the curing process, the wavelength of the FBG sensor was expected to shift back toward the original center wavelength. However, as illustrated in Figure 8, the temperature decreased to room temperature, the center wavelength measured using the FBG sensor shifted toward short wavelength. This measurement indicated that additional shrinkage occurred during the cool down of the resin, causing the initial residual strain in the interior of the FRP laminated plate.

### 3.2. Residual Strain Measurements of the Bottom Layer

This study used the FBG sensor to obtain the residual strain of VE resins with different stacking numbers and verified the relationship between the residual strain and stacking number.

Figure 9, Figure 10, Figure 11, Figure 12 and Figure 13 present the measurements of the FRP laminated plates with different stacking numbers. To enhance the clarity of the wavelength shift data, the center wavelength of the ordinate was represented as the wavelength variation. In terms of the residual strain calculations, we assumed that all of the wavelength data collected using the FBG sensor were affected by the environmental temperature. Therefore, during the calculation of the actual shrinkage strain of the laminated plate, Equation (2) was applied for calibration; the origin of the strain was the supposed strain of the resin after the reaction. The measurement curves calibrated using the temperature compensation are depicted in Figure 9b, Figure 10b, Figure 11b, Figure 12b and Figure 13b.

The experiment results demonstrated that when the stacking number was 4, the shrinkage residual strain was only 87.37 με. However, if the stacking number was increased to 10, the residual strain substantially increased to 450.92 με; the residual strain of the bottom layer of the laminated plate increased under the increased stacking numbers. The residual strain of the laminated plate with stacking numbers of 20, 30, and 40 were 511.553, 614.342, and 1001.446 με, respectively, and the correspondence was approximately 0.1%.

We analyzed the relationship between the residual strain of the bottom layer of the laminated plate and the stacking number, as presented in Figure 14. This study used LT-800/M-225 glass cloth. During fabrication of the laminated plates using VARTM, the thickness of a single laminate was approximately 0.92 mm. Therefore, laminated plates with a stacking number of 10 and 40 had a thickness of approximately 1 and 4 cm, respectively. The experiment results indicated that use of a thin laminated plate reduced the residual strain after the curing of the resin. However, if the stacking number was increased to 10, the residual strain of the resin became more evident. The residual strain did not increase notably under stacking numbers of 10 to 30 but, after 40, increased to almost double.

We used the curing curve of resin with different stacking numbers (Figure 10, Figure 11, Figure 12 and Figure 13) and speculated a relationship between the residual strain and highest exothermic temperature during curing. Figure 10a, Figure 11a and Figure 12a depict the curing curve for stacking numbers of 10, 20, and 30, respectively. The maximum wavelength variation of the resin under these three conditions after post-curing exothermic reaction was between 0.8 and 0.9 nm, indicating that under these conditions, the exothermic reactions of VE resin successfully dissipated heat during curing. The heat was not dissipated effectively under a stacking number of 40, perhaps because of the large number of layers. The heat accumulated at the surface of the mold, and the wavelength variation corresponding to the highest exothermic temperature reached 1.25 nm, as illustrated in Figure 13a. Therefore, we speculated that the temperature during the resin curing was a crucial factor affecting the residual strain after curing.

### 3.3. Residual Strain Measurements of the Symmetry Layer

According to the monitoring results of the thermocouple buried in the laminated plate, the highest exothermic temperature during resin curing occurred in the symmetry layer of the laminated plate. Therefore, we buried an FBG sensor in the symmetry layer of the symmetric laminate to monitor the residual strain in the area during the curing of the laminated plate.

The following presents a discussion on the experiment results of the symmetry layer. The laminated plate was thin with a stacking number of four, and, thus, the problems of high exothermic reactions or difficulty in heat dissipation did not occur during the curing of the laminated plate. According to the experiment measurements, the residual strain of the symmetry layer was only 71.51 με, which was similar to the residual strain of the surface layer (87.37 με). Therefore, the discussion of the experiment results excludes the results obtained under a stacking number of four.

Figure 15 depicts the residual strains under stacking numbers between 10 and 40. When the stacking number reached 10, the curing of resin exhibited more exothermic reactions and heat dissipation difficulty. The residual strain of the symmetry layer of the laminated plate rose to 694.264 με, which was higher than that of the surface layer (450.921 με). However, unlike the bottom layer, the residual strain of the symmetry layer did not change considerably when the stacking number was increased to 20, 30, and 40. The experiment results indicated that, despite the increased stacking number, the residual strain of the symmetry layer of the laminated plated decreased to between 665 and 801 με.

We reviewed the results of the measured residual strain of the symmetry and bottom layers of the laminated plates, as presented in Figure 16. The residual strain of the resin during curing was affected by both the temperature and stacking sequence. The experiment measurements verified that the exothermic temperature of each layer during the curing of FRP laminated plates was different. In general, the higher the exothermic temperature, the greater the residual strain after curing. However, if the symmetric laminate method is applied to the laminated plates, even if the highest exothermic temperature reactions occur at the symmetry layer of the laminated plate during curing, the greatest residual strain is maintained within a specific range.

Regarding the glass cloth and stacking sequence design used in this study, when the stacking number was no more than 40, the residual strain was maintained between 665 and 801 με during curing of the symmetry layer. Typical FRP structures rarely have a stacking number higher than 30. Therefore, under this condition, the largest residual strain of the FRP laminated plate often occurred in the symmetry layer. However, FRP marine structures are often designed with a stacking number greater than 40. Therefore, such designs must account for the residual strain of the surface layer, which is closer to the mold.

## 4. Conclusions

This study successfully employed FBG in VARTM and monitored the exothermic reaction of FRP laminated plates during curing; the FRP laminated plates were fabricated using VE resin with different stacking numbers. The residual strain after curing was also studied. The results verified that higher stacking numbers and areas in which heat did not dissipate easily often induced greater residual strain. The study results are detailed as follows:

Resin with a stacking number of four exhibited slight residual strain after curing; this residual strain can be ignored.When a stacking number was higher than 10, the exothermic temperature of the VE resin during curing notably increased, as did the residual strain of the FRP laminated plate. For example, in the bottom layer, which was closer to the mold, the residual strain slowly increased as the stacking number increased from 10 to 30; the residual strain was approximately between 450.9 and 614.3 με. When the stacking number was increased to 40, the heat dissipation problem caused the residual strain of the bottom layer to increase to 1004.45 με. The study results indicated that if VE resin is used in thick laminated structures, the heat dissipation of the mold must be considered.The residual strain of the laminated plate was affected by the exothermic temperature of the resin during curing as well as the stacking sequence. For example, in the symmetry layer of the laminated plate, the temperature distribution was measured at different thicknesses, revealing the symmetry layer to have the highest average temperature. However, the glass cloth and stacking sequence design used in this study resulted in the symmetric laminate of the symmetry layer maintaining the residual strain within a range of 665 to 801 με
The testing methods and results of this study can be referenced for FRP marine structure design. Multichannel measurements can be applied in the future, such as the simultaneous monitoring of the residual strain distribution of FRP laminated plates with different stacking numbers, to serve as a reference for structural design and assessment of structural safety.

## Figures and Tables

**Figure 1 polymers-14-01446-f001:**
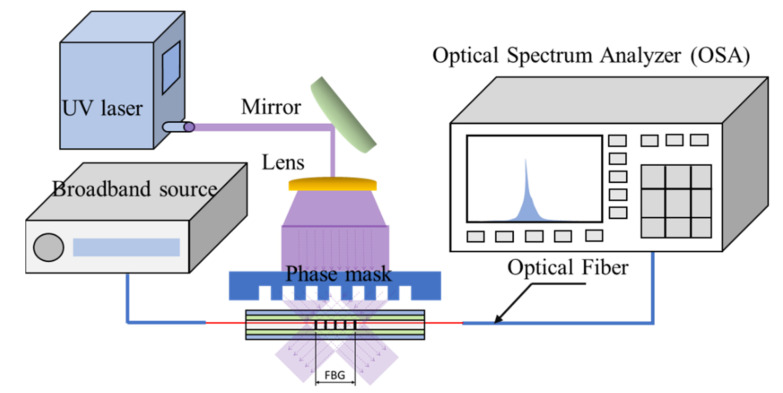
Using an ultraviolet laser to fabricate the FBG sensor.

**Figure 2 polymers-14-01446-f002:**
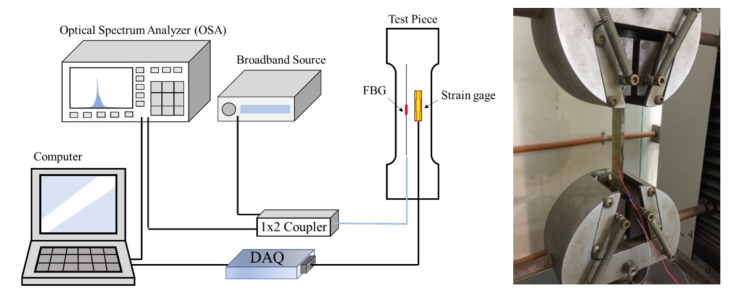
Schematic diagram of relationship between the wavelength shift and strain of the FBG sensor in the tensile test.

**Figure 3 polymers-14-01446-f003:**
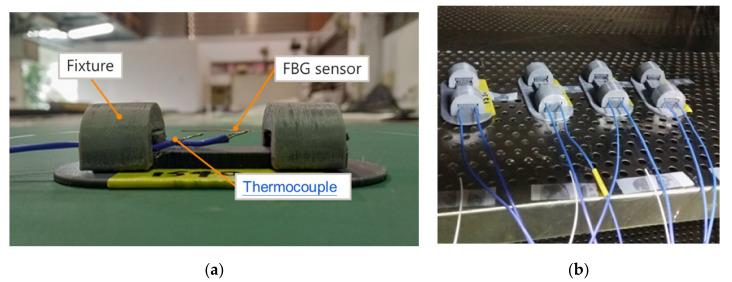
FBG and thermocouple sensor. (**a**) Fixing method of FBG sensor and thermocouple. (**b**) Schematic diagram of heating test.

**Figure 4 polymers-14-01446-f004:**
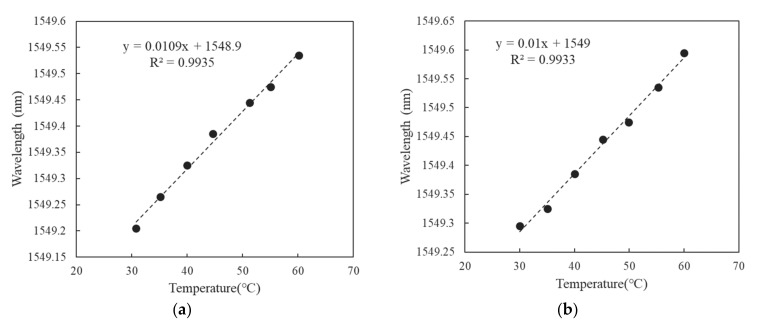
Wavelength and temperature curve. (**a**) Test result of the first set. (**b**) Test result of the second set.

**Figure 5 polymers-14-01446-f005:**
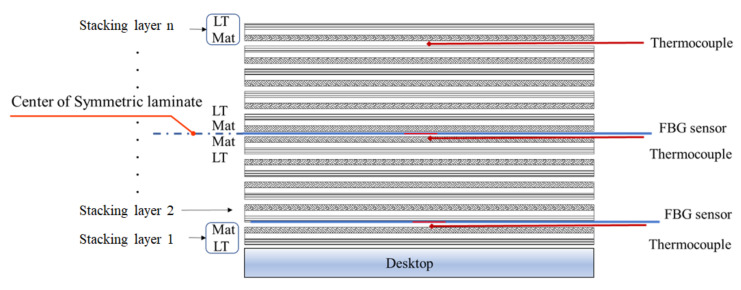
Symmetrical stacking sequence and sensor placement.

**Figure 6 polymers-14-01446-f006:**
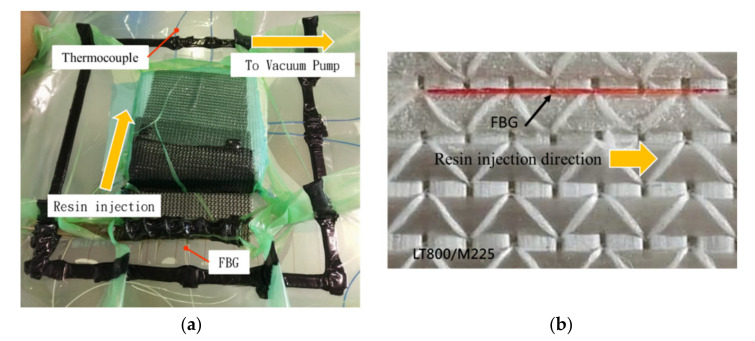
FRP test piece injection and FBG sensor placement. (**a**) VARTM experiment setup. (**b**) Resin injection direction and FBG sensor placement.

**Figure 7 polymers-14-01446-f007:**
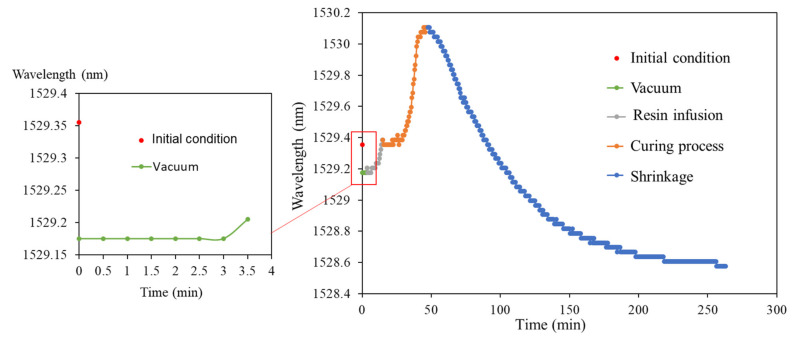
Measurements of the curing of FRP with 30 layers.

**Figure 8 polymers-14-01446-f008:**
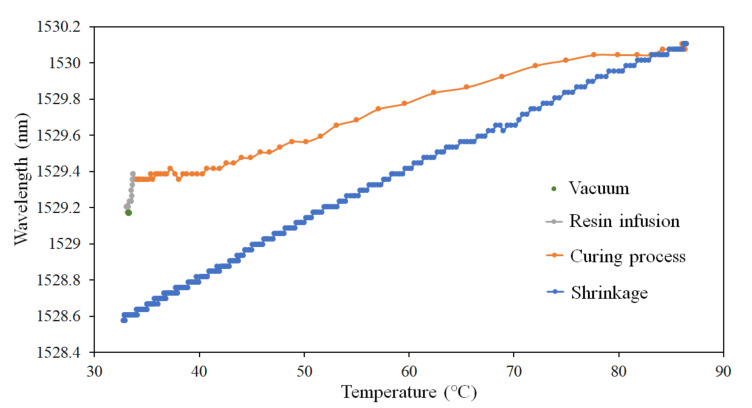
Wavelength shift and temperature change relationship curve.

**Figure 9 polymers-14-01446-f009:**
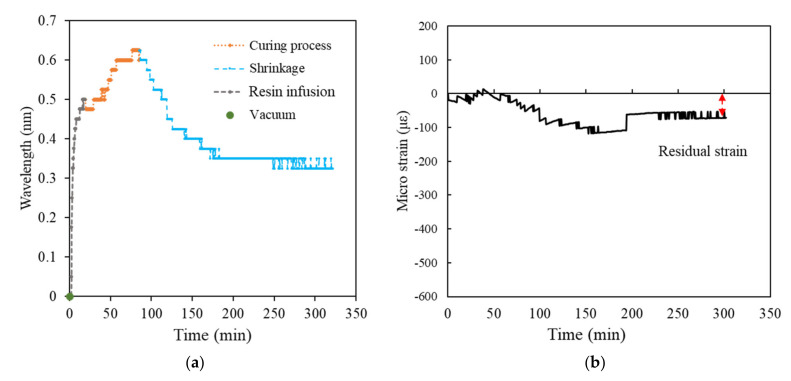
Experimental results of surface laminate-stacking number is 4. (**a**) Wavelength-variation varying with time. (**b**) Residual strain.

**Figure 10 polymers-14-01446-f010:**
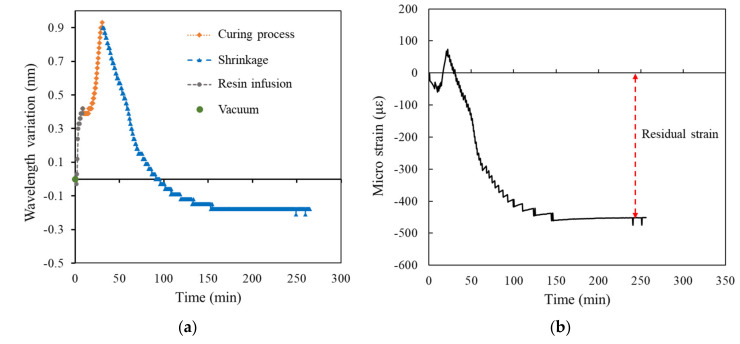
Experimental results of surface laminate-stacking number is 10. (**a**) Wavelength-variation varying with time. (**b**) Residual strain.

**Figure 11 polymers-14-01446-f011:**
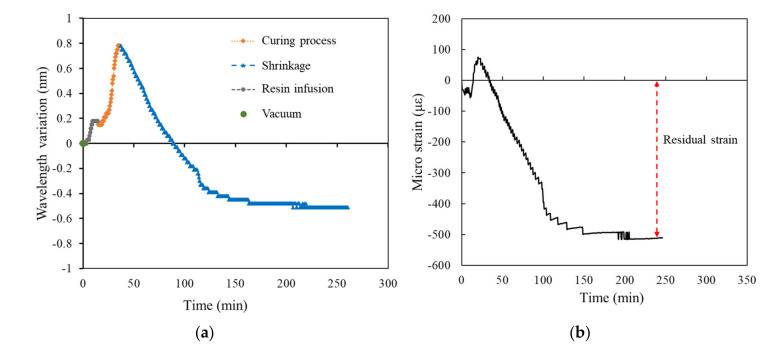
Experimental results of surface laminate-stacking number is 20. (**a**) Wavelength-variation varying with time. (**b**) Residual strain.

**Figure 12 polymers-14-01446-f012:**
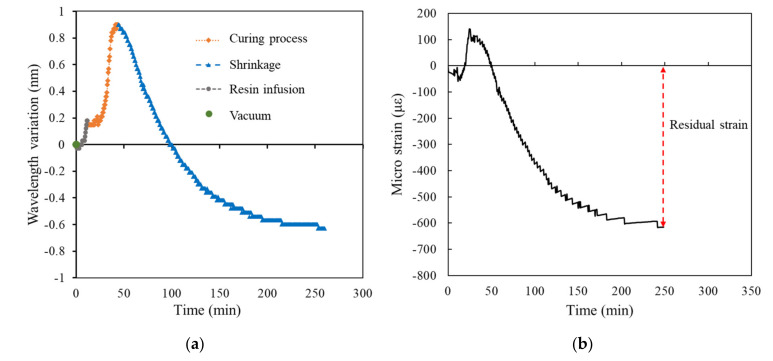
Experimental results of surface laminate-stacking number is 30. (**a**) Wavelength-variation varying with time. (**b**) Residual strain.

**Figure 13 polymers-14-01446-f013:**
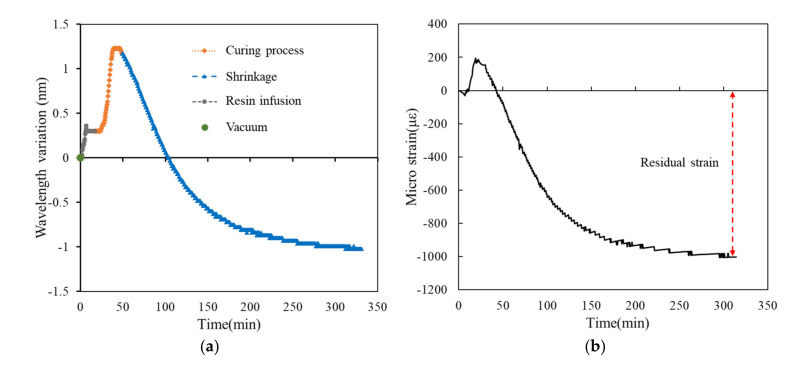
Experimental results of surface laminate—stacking number is 40. (**a**) Wavelength-variation varying with time. (**b**) Residual strain.

**Figure 14 polymers-14-01446-f014:**
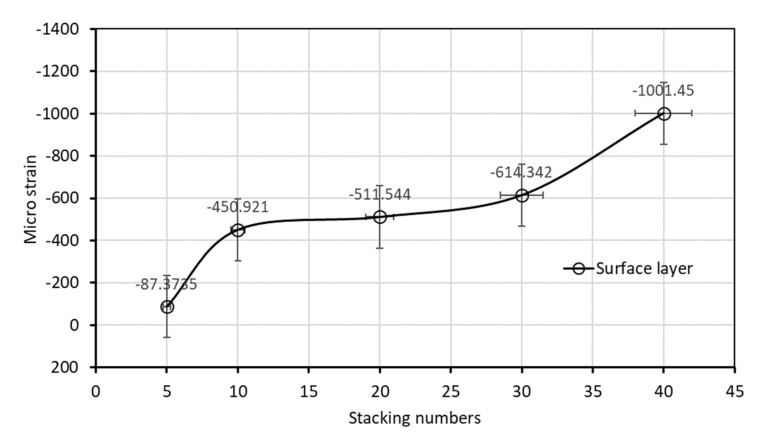
Residual strains of surface laminate.

**Figure 15 polymers-14-01446-f015:**
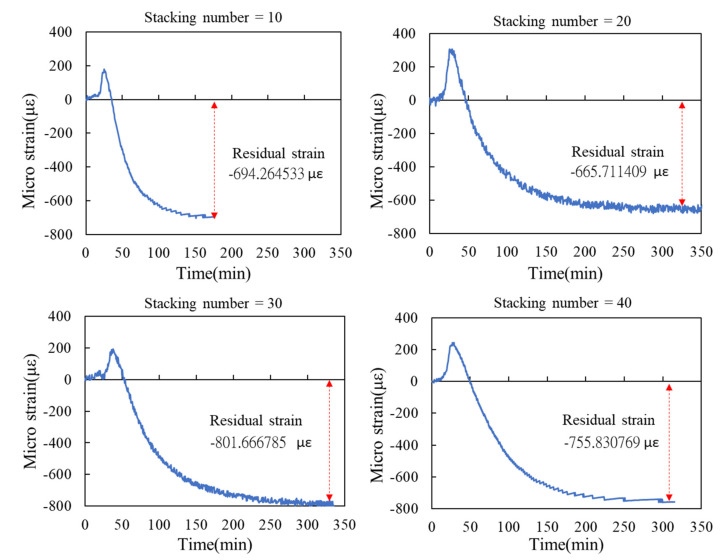
Residual strains of symmetrical layer.

**Figure 16 polymers-14-01446-f016:**
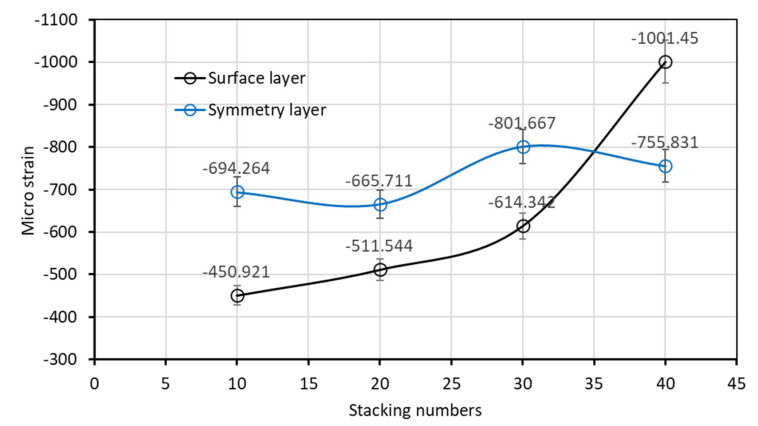
Residual strain measurement of the bottom layer and symmetry layer of the laminated plate.

**Table 1 polymers-14-01446-t001:** Relationship between the wavelength shift and strain of the FBG sensor.

**Time (S)**	**(A)**	**(B**)	**(C) = (B/A)**	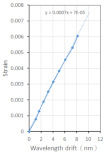
	**Wavelength Shift (nm)**	**Strain**	**Strain Corresponding to a Shift of 1 nm**	
0	0	0	0	
20	1.118	$7.841\times 10^{-4}$	$7.013\times 10^{-4}$	
40	1.677	$1.288\times 10^{-3}$	$7.680\times 10^{-4}$	
60	2.395	$1.891\times 10^{-3}$	$7.896\times 10^{-4}$	
80	3.194	$2.516\times 10^{-3}$	$7.877\times 10^{-4}$	
100	4.032	$3.156\times 10^{-3}$	$7.827\times 10^{-4}$	
120	5.030	$3.835\times 10^{-3}$	$7.624\times 10^{-4}$	
140	6.088	$4.550\times 10^{-3}$	$7.474\times 10^{-4}$	
160	7.285	$5.301\times 10^{-3}$	$7.275\times 10^{-4}$	
180	8.164	$6.050\times 10^{-3}$	$7.411\times 10^{-4}$	
Average			$7.564\times 10^{-4}$	

## Data Availability

Data is contained within the article.

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
