# Peer review of "Using a Fiber Bragg Grating Sensor to Measure Residual Strain in the Vacuum-Assisted Resin Transfer Molding Process"

_polymers, 2022, doi:10.3390/polym14071446_

Round 1

Reviewer 1 Report

In this paper, the authors explored the exothermic reaction and shrinkage of VE resin and glass fiber during the vacuum-assisted resin transfer molding process, as measured using a fiber Bragg grating sensor. The experiment results verified the relationship between the stacking number and residual strain shrinkage. In addition, the symmetric laminate method was used to prevent the bending–twisting coupling effect and subsequent warping deformation of the FRP laminated plate during curing. Several suggestions are supplied:
1. Suggest the authors give more detail about the FBG fabrication setup, such as the wavelength of the laser system and the period of the phase grating, etc.
2. Figure 2 presents one schematic of the experiment, not the relationship between wavelength shift and strain, pls check.
3. Suggest the authors make the equations uniform, pls check the PDF version. 
4. Suggest making the graphs in section 3.2 more focused.
By the way, the methods present here also have some potential applications in marine environment monitoring, suggest the authors introduce one latest review on this topic:
Optical fiber sensing for marine environment and marine structural health monitoring: A review Optics and Laser Technology, 2021.

Author Response

  1. The OSA used in this study is Anritsu MS9710C, the measurement range of spectral wavelength is 600~1750nm, and the maximum spectral resolution can be accurate to 0.05nm. In order to prevent the waveforms measured by each stacking layer from overlapping, we adjusted the spacing of phase mask to obtain different center wavelengths. The effective refractive index of optical fiber used in this study is 1.468, and the spacing of phase mask considered in this study is between 0.521~0.531. Finally, we can obtain the optical fiber with center wavelength of 1530~1560nm. Added instructions have been corrected in the article.
  2. The title of Figure 2 has been revised to "Schematic diagram of relationship between the wavelength shift and strain of the FBG sensor in the tensile test."
  3. It should be the format error when converting WORD to PDF, and the equation format of the original WORD file is correct.

  4. Thanks for the reviewer's suggestion. Because the wavelength shift of laminates with different stacking numbers is different, we use multiple graphs to present the test results truthfully.

    The article suggested by the reviewer has been added to Ref. 7.

Reviewer 2 Report

Dear authors,

your work is very interesting and I think it can be published after a few improvements:

  1. Please, give an e-mail for correspong author.
  2. The font in Abstract changed after 'In addition,...'
  3. Lines 155-156 'phase mask' is more usual than 'phase grating'
  4. Line 157 - a think that structure was created 'in' optical fiber
  5. Line 165 - please, check the Ref.15, as I undertand it is not about FBG and does not contain information about wavelength shift.
  6. And the most important remark:

You didn't explain the algorithm of peak wavelength determination. Based on Fig.7-8 I suppose you take simple maximum value of optical spectrum. So you have resolution ~0.03 nm. It causes quantization in all plots of deformation measurements and 'jumps' in plots 9-13,15, which are not real and physical. As your residual strain is quite big in relation to this it will not affect a lot on result. But for understanding of process it is needed to mention the peak determination algorithm and its influence. If you will determine the peak as a mass center for some ~5-15 points - you will get more noise between neighbouring measurements but less jumps. Also showing deformation values down to 0.001 με (line 337) is not true in such conditions. For figures 14 and 16 it will be useful to show error bars.

If you have not only values but all saved spectra - it can be a great improvement to recalculate peak values with mass center algorithm.

Author Response

  1. The corresponding author's E-MAIL has been added.
  2. It should be a format error caused by the file conversion. The original WORD file format is correct.

  3. Thank you for your reminder. Description has been revised.
  4. Thank you for your reminder. Description has been revised.
  5. Thank you for your reminder. Description has been revised.
  6. Indeed, we directly take simple maximum value of optical spectrum, and the sampling frequency is 30 seconds. We have added instructions to the manuscript.

    Furthermore, the strain in line 337 is a wrong description and has been removed.

Reviewer 3 Report

This manuscript describes the use of fiber Bragg grating sensors to measure residual strain in the vacuum-assisted resin transfer molding process. This is an interesting work with a considerable impact on manufacturing applications, particularly for the maritime industry. It includes a full description of the fiber grating characterization, as well as all the information needed to replicate the setup of the sensors in a similar scenario. The results are very relevant, since they allow to deeply study the strain shrinkage difference between the symmetry layer and the surface layer. Additionally, the paper is well written, making a proper use of the English language. It features a lot of introductory material, as well as relevant figures illustrating each step of the work. And most importantly, the Conclusions are perfectly aligned with the obtained experimental results. All in all, this is a very nice piece of work and a perfect match for Polymers. I will just recommend the following minor additions to be made before its acceptance for publication:

 - The fifth paragraph of the Introduction provides some examples of applications of FBG sensors for temperature monitoring. I suggest to add this work: Sensors and Actuators A: Physical 332, 113061 (2021), which demonstrates how FBG sensors are embedded into heating surfaces for continuous temperature monitoring.

 - In Section 2.1, the authors mention that they use an optical fiber with a diameter of 245 µm. When they describe the FBG manufacturing process, they do not mention whether they strip the fiber or not, so it is not clear if the coating diameter is 245 µm or if the acrylate diameter is 245 µm. I guess it is the first case, since they mention that it is a single-mode fiber, but this should be specified.

 - In table 1, all the results belonging to the same column should have the same number of decimal places for the sake of precision. Additionally, a graph could be included with the strain evolution as a function of the wavelength shift.

 - In Section 3.1, could the authors discuss if differentiating the strain and temperature effects could be beneficial for the present work? I mean something like the work in Applied Optics 58 (18), 4898-4904 (2019). Could the authors perform a similar approach in this case? Would be an added value?

Author Response

  1. The article suggested by the reviewer has been added to Ref. 27.
  2. The optical fiber used in this study is Corning SMF-28e, and the coating diameter is 245.

    We have added instructions to the manuscript.
  3. Thank you for your reminder.

     All decimal places have been unified in table 1, and the graph of strain versus wavelength shift has also been added in table 1.
  4. The multi-layer coating technology used by Yard can indeed solve the problem of this study. However, there is no such energy and equipment in our laboratory. Yard's method has been presented in the literature.

Round 2

Reviewer 1 Report

The authors improved the draft with the comments from the reviewers, so now I can recommend it publish on Polymers.

Reviewer 2 Report

Thank you for your answers.

Suggest to use improved peak detection algorithm in further experiments.